# The Effect of Different Morphologies of WO_3_/GO Nanocomposite on Photocatalytic Performance

**DOI:** 10.3390/ma15228019

**Published:** 2022-11-14

**Authors:** Banu Esencan Türkaslan, Aziz Kerim Çelik, Ayça Dalbeyler, Nicholas Fantuzzi

**Affiliations:** 1Department of Chemical Engineering, Faculty of Engineering, University of Süleyman Demirel, Isparta 32260, Turkey; 2Department of Civil, Chemical, Environmental, and Materials Engineering, University of Bologna, 40126 Bologna, Italy

**Keywords:** tungsten trioxide, graphene oxide, photocatalysis, nanocomposite

## Abstract

Tungsten trioxide/graphene oxide (WO_3_/GO) nanocomposites have been successfully synthesized using in situ and ex situ chemical approaches. Graphite and tungsten carbide (WC) were employed to perform in situ synthesis, and WO_3_ and GO were employed to perform the ex situ synthesis of WO_3_/GO nanocomposites. GO, which was required for ex situ synthesis, is synthesized via the modified and improved Hummers method. XRD, SEM/EDS, and FTIR are used for the characterization of the nanocomposite. From the XRD of the WO_3_/GO nanocomposites, it was observed that WO_3_ distributed uniformly on graphene oxide sheets or was incorporated between the sheets. The photocatalytic activities of WO_3_/GO nanocomposites were evaluated by methylene blue (MB) adsorption and visible light photocatalytic degradation activities by UV-vis spectroscopy. The results showed that the efficiency of the photocatalytic activity of the nanocomposite depends on different synthesis methods and the morphology resulting from the changed method. WO_3_/GO nanocomposites synthesized by both methods exhibited much higher photocatalytic efficiencies than pure WO_3_, and the best degradation efficiencies for MB was 96.30% for the WO_3_/GO in situ synthesis nanocomposite.

## 1. Introduction

In recent years, due to a notable increase in industrial activities that produce wastewater, the development of easily accessible adsorbent materials for cleaning water resources gained great momentum. Organic dyes have a dangerous and toxic effect on humans, which are found in the wastewater of various industries [1].

Among the different methods, degradation by photocatalysis has become a promising technique as it is simple and sustainable, and it enables the conversion of dyes to non-hazardous waste.

Nanocomposite materials, including metal oxide nanoparticles, with their advanced physicochemical properties, are common photoactive semiconductor materials that make them suitable for use in photocatalysis [2,3].

Tungsten oxide (WO_3_), a nanostructured metal oxide, is widely used in the fields of gas detection [4,5], lithium-ion batteries [6,7], smart windows [8], and photocatalysis [9,10] practices. However, minimal photonic efficiency, the fast recombination rate of the charge carriers (photogenerated hole–electron), and low absorption ranges restrict the performance of WO_3_ as an efficient photocatalyst [11,12,13,14]. The most important features that distinguish an ideal adsorbent from others are high adsorption capacity, rapid adsorption rate, and high selectivity. In this sense, doping WO_3_ with other elements or compounds to improve its photocatalytic activity is deemed as an effective approach [15,16,17]. This may help improve the morphology of WO_3_ material as well as the affected band levels and characteristics of the charge carriers [18]. 

In particular, graphene and its derivatives are frequently preferred in improving the performance of metal oxide nanostructures due to their extraordinary properties [19,20,21]. 

Contrary to graphene, the use of hydrophilic GO, which contains various proportions of carbon, oxygen, and hydrogen in its structure, is expanding day by day due to its easy dispersion in solutions, its dielectric properties, transparency and adjustable electronic properties [22,23]. However, it is difficult to separate GO from water after adsorption. To overcome this drawback, the hybridization of GO with other inorganic or organic materials is an alternative [24].

Hummers and Offeman synthesized GO via the method of oxidizing graphite in H_2_SO_4_, NaNO_3_, and KMnO_4_ atmospheres in 1958 [25]. Hummers’ methodology was widely accepted, yet many disadvantages of this method have been reported, such as the production of toxic gas (NO_2_ and N_2_O_4_), the residual nitrate, and low efficiency levels. In the last 20 years, alternative methods have been attempted relative to Hummer’s method, including the addition of a peroxidation phase before KMnO_4_ oxidation (without NaNO_3_), increasing the amount of KMnO_4_ rather than NaNO_3_, and replacing KMnO_4_ with K_2_FeO_4_ when NaNO_3_ is extracted [26,27].

Studies conducted on WO_3_/GO show that while the photocatalytic degradation activities of composites formed by combining WO_3_ and GO structures that are synthesized separately via the ex situ method [28,29,30,31,32], there are no studies conducted regarding the photocatalytic degradation activities of nanocomposite structures, which are synthesized via in situ methods. It is thought that the composite structures synthesized using different methods are in the form of nanoparticles and nanowires [33,34,35,36], and this will affect the photocatalytic efficiency level.

In this study, highly efficient photocatalysts were developed, which is a method of producing renewable energy. To this end, both ex situ and in situ syntheses of WO_3_/GO composites were performed, and the photocatalytic activity of the formed structure was examined. GO was synthesized via the modified and improved Hummers method without using NaNO_3_. Given that there is no other WO_3_/GO study conducted to examine the photocatalytic degradation activity of nanocomposite structures synthesized via in situ methods, our study bears the characteristics of the first study conducted in this field, and it is also the first study that compares photocatalytic activities of ex-situ and in-situ synthesized WO_3_/GO composite structures.

## 2. Materials and Methods

Graphite flake (≥75% min), sulfuric acid (H_2_SO_4_, 98%), potassium permanganate (KMnO_4_, 99%), hydrogen peroxide (H_2_O_2_, 30%), hydrochloric acid (HCI, 37%), and tungsten (VI) oxide (WO_3_, <100 nm) were obtained from Sigma-Aldrich, and tungsten carbide (WC 45 nm, 99%) was obtained for the synthesis process from Nanokar company (İstanbul, Turkey).

### 2.1. Graphene Oxide (GO) Synthesis

GO was synthesized from layered graphite via the modified and improved Hummers method. Firstly, graphite (2 g) and then KMnO_4_ (6 g) were gradually added into H_2_SO_4_ within the ice bath and mixed. Later, 300 mL of deionized water was added to the mixture. In order to stop the oxidation process and remove the impurities in the structure, the mixture was filtered by adding H_2_O_2_ and HCl, respectively. Synthesized graphite oxide measuring 1 g was taken into 350 mL of pure water and dispersed for 3 h. After two hours of sonication in order to facilitate the exfoliation of clumped graphite oxide layers on GO layers, the mixture was centrifuged, and GO was produced.

### 2.2. In Situ and Ex Situ Synthesis of WO_3_/GO Composites

A total of 10 mg WO_3_ and 30 mg GO were separately mixed for 2 h in 10 mL and 30 mL water, respectively, for the ex situ synthesis of WO_3_/GO nanocomposites. Later, the two solutions were combined and mixed for another 2 h in a magnetic stirrer. The achieved solution was centrifuged and finally dried at 60 °C for 24 h to obtain the WO_3_/GO composite.

WC powder measuring 1 g and 10 mL H_2_SO_4_ were mixed in ice bath for the in situ synthesis of WO_3_/GO nanocomposite. KMnO_4_ measuring 3 g was gradually added into this mixture by continuously stirring. After adding KMnO_4_, the mixture was stirred for an additional 2 h; then, 15 mL H_2_O_2_ (%30 *w*/*w*) was added, and it was observed that the color of the mixture turned bright yellow. The achieved solution was centrifuged and finally dried at 60 °C for 24 h.

### 2.3. Photocatalytic Activities of WO_3_/GO Composites

Methylene blue (MB) was used as typical pollutants to study the photocatalytic activity of the synthesized WO_3_/GO composites, which are synthesized with in situ and ex situ methods. In order to simulate the coloring agent, 75 mL of 20 ppm MB solutions was prepared. WO_3_ was added to one of the solutions, and 15 mg of in situ WO_3_/GO and 15 mg of ex situ WO_3_/GO were added to other solutions as catalysts. Then, the solution was deposited into tubes in equal amounts in order to be able to make measurements at different time intervals.

After mixing for 30 min in the dark to ensure an adsorption–desorption balance, the solution tubes that were placed in the UV cabinet were exposed to a total of 2 xenon lamps, each of which was 150 Watt (Figure 1).

The distance between the lamp and the center of the tubes was measured as 8 cm. Samples were then taken at regular intervals to observe the degradation of methylene blue at 660 nm. The first measurement was made in the 15th minute; the next measurements were set to be made every 30 min, and regular measurements were made. 

## 3. Results

The GO characterization was performed with X-ray diffraction (XRD) and the scanning electron microscopy (SEM/EDS) technique. WO_3_/GO composites were also evaluated by using a scanning electron microscope (SEM, Quanta Feg 250; FEI, Eindhoven, the Netherlands). WO_3_/GO composites were examined with a low-vacuum at 20.00 kV and 12.7–13.2 mm working distance at 5000× and 10000× magnifications, respectively. The elemental analysis of nanocomposites was carried out using an SEM microscope equipped with an energy-dispersive X-ray spectroscopy (EDX, Quanta Feg 250; FEI, Eindhoven, The Netherlands). The distribution and atomic composition of WO_3_/GO was examined using elemental mappings at an accelerating voltage of 20 kV. The crystalline phase of WO_3_/GO were examined by X-ray diffraction (XRD, Bruker D8 Advance Twin-Twin; Bruker, Karlsruhe, Germany) at 40 kV, 40 mA, and 1600 watts. In order to determine the photocatalytic capacity of WO_3_/GO nanocomposites, a UV-Vis spectrophotometer (UV-Vis Carry 60) device was used.

### 3.1. Characterization of GO

When the XRD spectrum of the GO structure was examined, it was observed that the peak formed at 2θ = 11.52° for GO, which is consistent with the results in the literature and it shows that the GO structure is obtained properly (Figure 2) [37].

SEM analysis results showed that the GO structure was formed by layered wavy structures piled on top of each other (Figure 3). 

### 3.2. Characterization of WO_3_/GO Composites

Figure 4 provides the comparative FTIR spectra of GO and WO_3_/GO composites. The peaks of <1000 cm^−1^ within the composite structures show the existence of pure WO_3_ [28]. 

The FTIR spectrum of GO shows that the hydroxyl bond (-OH) is at 3425 cm^−1^, the carbonyl bond (C=O) is at 1719 cm^−1^, the aromatic bond (C=C) is at 1627cm^−1^, the epoxy bond is at 1627 cm^−1^, (C-O) is at 1400 cm^−1^, and the alkoxy bond (C-O) is at 1064 cm^−1^. The bands at 1715 cm^−1^ and 1617 cm^−1^ in the in situ synthesized composite structure and the bands at 1713 cm^−1^ and 1614 cm^−1^ in the ex situ synthesized composite structure belong to C=O and C=O vibrations, indicating the presence of GO in the composite’s structure.

Peaks under <1000 cm^−1^ observed in composites that are not observed in GO structure are caused by O-W-O stretch vibrations and show that nanoparticles bind to GO nanolayers strongly [38]. O-W-O vibrations observed approximately at 820 and 758 cm^−1^ in the in situ synthesized composite structure were observed less in ex situ structures.

SEM micrographs were used to identify the morphology of the synthesized composites and the location of the metal oxide in the carbon matrix. It was observed that while the pure WO_3_ structure comprised spongy structures of various sizes placed in such a way as to form spaces between them, WC consisted of randomly distributed and irregularly shaped coarse grains (Figure 5).

It is observed that the wavy interlayer spaces in GO structures are randomly dispersed by some spherical WO_3_ particles to form smoother surfaces in the WO_3_/GO ex situ synthesized nanocomposite (Figure 5e,f) structures. In WO_3_/GO insitu synthesis morphologies, however, it is seen that a single-phase homogeneous composite morphology formed with good interfacial interaction between GO and WC (Figure 5g,h). The homogeneous coating of the GO surface with WO_3_ as a result of good interfacial interaction in the in situ synthesis structure shows parallelisms with the O-W-O vibration bands observed in the FTIR results, while the lesser observation of these bands in the ex situ synthesis also supports the WO_3_ particle structure observed between the GO layers.

The EDS analysis of WO_3_ powders shows a tungsten atomic percentage of 79.35% and an oxygen atomic percentage of 20.65%. WC powders show tungsten at 90.34% and a carbon atomic percentage of 9.66%. After the synthesis of nanocomposites, a notable decrease in the atomic percentages of W elements was observed. Furthermore, the presence of GO was confirmed via EDS analyses, which showed additional carbon elements and oxygen elements in WO_3_/GO ex situ and in situ, respectively (Table 1).

Figure 6 shows the XRD patterns of the ex situ and in situ WO_3_/GO, WO_3_/GO nanocomposites, which confirms the presence of both WO_3_ and GO.

A low intensity peak at 10.8° indicates the formation of GO sheets in the in situ synthesis of WO_3_/GO, which is due to the poor crystalline nature of carbon. The other peaks at 2θ values of 23.3, 24.5, 34.2, 42.1, 47.49, and 50.22 confirm the presence of WO_3_ particles [39].

The WC diffraction spectrum shows three major intense peaks located at 2θ = 30.42°, 38.98°, and 47.03°, which correspond well to the crystallographic planes (001, 100, and 101) of WC, respectively [40].

### 3.3. Photocatalytic Degradation

Figure 7 shows the adsorption capacities (q_e_) of WO_3_ and WO_3_/GO ex situ and WO_3_/GO in situ nanocomposites, which were calculated using the following formula:$$q_{e}=\frac{C_{0\;}-C_{e\;}}{m}*V$$
where *C*_0_ (mg/L) and *C_e_* (mg/L) refer, respectively, to the initial concentration of the coloring agent and the concentration of the coloring agent remaining in the solution after adsorption, *m* (g) refers to the amount of adsorbents, and V (mL) represents the volume of the solution. Accordingly, WO_3_ showed adsorption capacities in the range of 12.65–15.59, while WO_3_/GO ex situ and WO_3_/GO in situ nanocomposites showed adsorption capacities in the range of 15.18–19.84 and 19.46–23.91, respectively. During the experiment, the adsorption maximum capacity was determined with WO_3_/GO in situ nanocomposites at t = 270, and the lowest was determined with WO_3_ at t = 0. Within the increasing time intervals, the adsorption capacity of WO_3_/GO in situ nanocomposites showed a significant increase after 3 h. This shows that this situation can be associated with the surface area of the nanocomposite, thus leading to the understanding that the photocatalytic effect increases with time.

The degradation efficiency of was calculated using the following formula:$$\eta \%=(1-\frac{C}{C_{0}})\times 100$$
where *C*_0_ is the absorption maximum at t = 0, and *C* is the absorption maximum after complete degradation.

WO_3_ and WO_3_/GO nanocomposites synthesized ex situ/in situ showed the maximum degradation of 75.79%, 90.52%, and 96.30% respectively (Figure 8). It was observed that the degradation amount of the WO_3_/GO in situ catalyst was higher than the one of ex situ synthesis, and it was concluded that this difference depends on the synthesis method. When GO is added to the matrix, it increased the photocatalytic effect, and in parallel with this, the synthesized nanocomposite structures reach higher percentage degradation efficiency values in a shorter time compared to WO_3_.

The chemical structure of MB has cationic atoms and aromatic rings. The degradation mechanism starts with the MB dye adsorption on the nanocomposite’s surface followed by its photodegradation, which can be summarized in the following steps. Figure 9 illustrates the mechanism of the photocatalytic degradation of MB via the WO_3_/GO nanocomposite’s UV irradiation. First, visible light radiation allows the transfer of electrons in in the valence band WO_3_ to the conduction band of GO. Therefore, holes (h) and electrons (e-) are formed on the surface of the WO_3_ photocatalyst. GO behaves as an electron acceptor via electrostatic and π−π stacking interactions. Then, while the holes react with the hydroxide ion, the electrons react with dissolved oxygen to produce OH^-^, which degrades MB dyes into non-toxic gases such as carbon dioxide and water. In addition, hydrogen peroxide reacts with electrons to produce more OH- to increase the degradation of the dye.

As shown in Table 2, prepared photocatalysts were compared with other WO_3_-based nanocomposites. Their photocatalytic efficiency is usually at 80–97%. In this work, WO_3_/GO nanocomposites prepared by in situ synthesis were higher than that of the tungsten-oxide-based counterparts. In addition, the method is simple and does not require synthesizing GO separately; the in situ oxidation of graphite is provided while the nanocomposite is formed.

## 4. Discussion

The photocatalytic activities of the WO_3_ and WO_3_/GO nanocomposites were evaluated by the degradation of MB in aqueous solutions. Compared with WO_3_, the photocatalytic activity of the WO_3_/GO nanocomposites was enhanced, and the best degradation efficiencies for MB was 96.30% for the WO_3_/GO in situ synthesis nanocomposite. It was attributed to the large surface area of GO, which served as an acceptor of the electrons generated in the WO_3_ and effectively decreased the recombination. To overcome the rapid recombination and slow migration of charge carriers, different morphologies have been developed, such as nanoplates, nanotubes, and nano-sheets. The homogeneous nanoplate structure formed in in situ syntheses showed higher photocatalytic effects due to its large surface area than the nanorod-like structure formed as a result of ex situ syntheses.

Although there are various different pollutant sources in the environment [42], the application of composite-based photocatalysts is limited to water treatments. Expanding the application areas of GO-based photocatalysts with different studies is necessary. Laboratory equipment is mostly used for the degradation processes in the laboratory. To prepare photocatalysts on a commercial scale, cost studies should be carried out for the large-scale degradation of pollutants, and systems should be modified with appropriate strategies. It is seen that suitable morphologies can be obtained by adjusting different methods and reaction conditions, and morphology control is an important parameter for photocatalytic activities.

## 5. Conclusions

In summary, we reported a simple chemical in situ and ex situ synthesis process and the physical properties and photocatalytic activities of WO_3_/GO composite structures. We observed that different synthesis methods affect WO_3_/GO’s morphology, while different WO_3_/GO morphologies affect photocatalytic performances. The ex situ preparation of the composite leads to the formation of well-dispersed WO_3_ with smoother surface in the WO_3_/GO. However, the in situ-prepared WO_3_ nanostructures have showed that single-phase homogeneous composite morphologies formed with good interfacial interactions between GO and WC. The prepared WO_3_ and its nanocomposite with GO was evidenced for the dye degradation of MB. The best degradation efficiencies for MB were 96.30% for the WO_3_/GO in situ synthesis nanocomposite, which are much better than that of WO_3_. The results showed that WO_3_/GO composites exhibited an enhanced WO_3_ photocatalysis efficiency in visible light. This study gave a new perspective for applications of WO_3_/GO nanocomposite photocatalysts for various areas.

## Figures and Tables

**Figure 1 materials-15-08019-f001:**
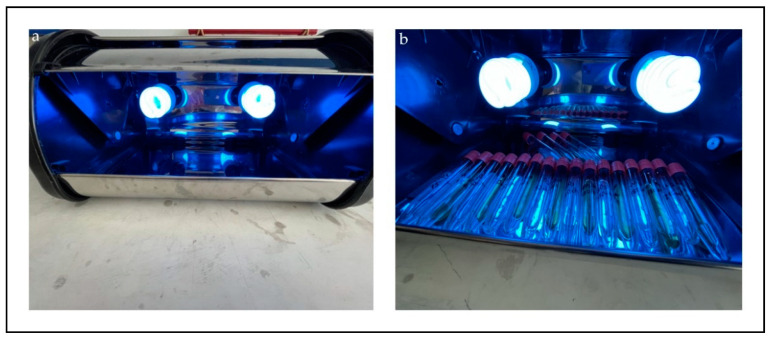
Designed UV cabinet (**a**) and tubes prepared for measurements to be made with different time intervals (**b**).

**Figure 2 materials-15-08019-f002:**
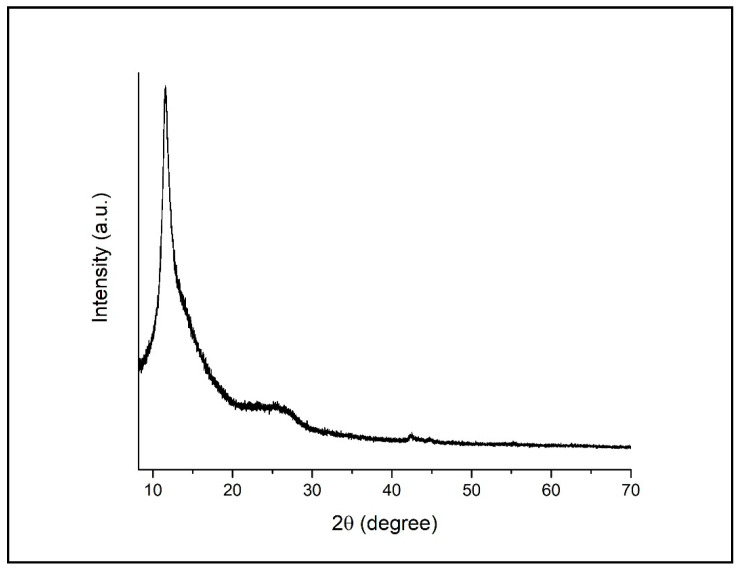
XRD patterns of GO.

**Figure 3 materials-15-08019-f003:**
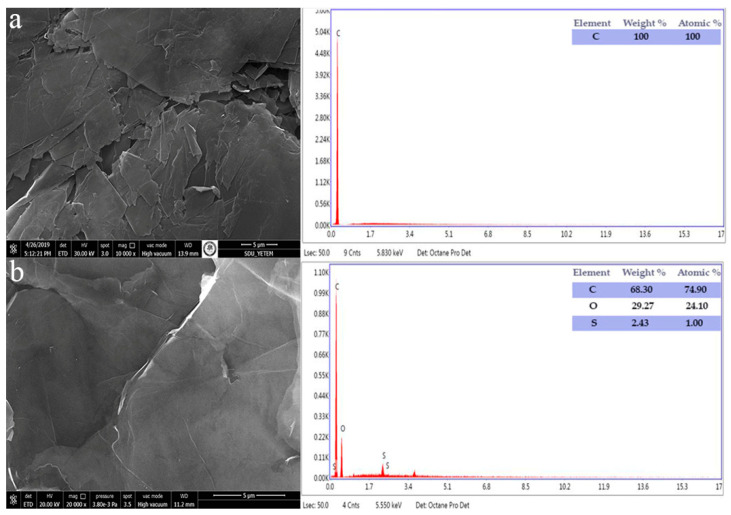
SEM/EDS image of (**a**) graphite and (**b**) GO.

**Figure 4 materials-15-08019-f004:**
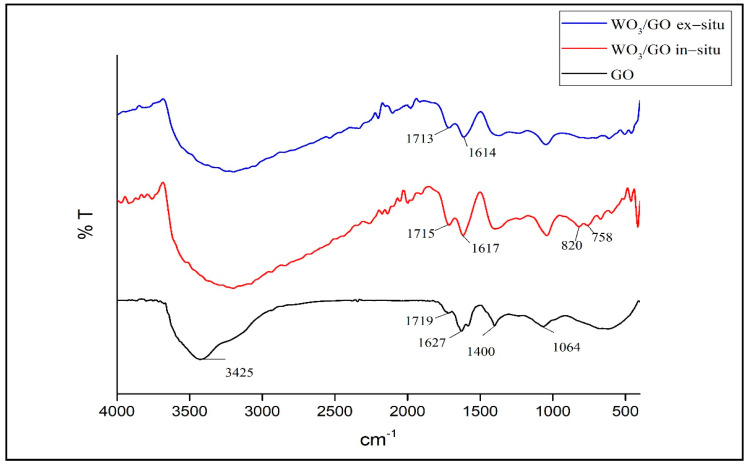
FTIR Spectra of GO and WO_3_/GO Composites.

**Figure 5 materials-15-08019-f005:**
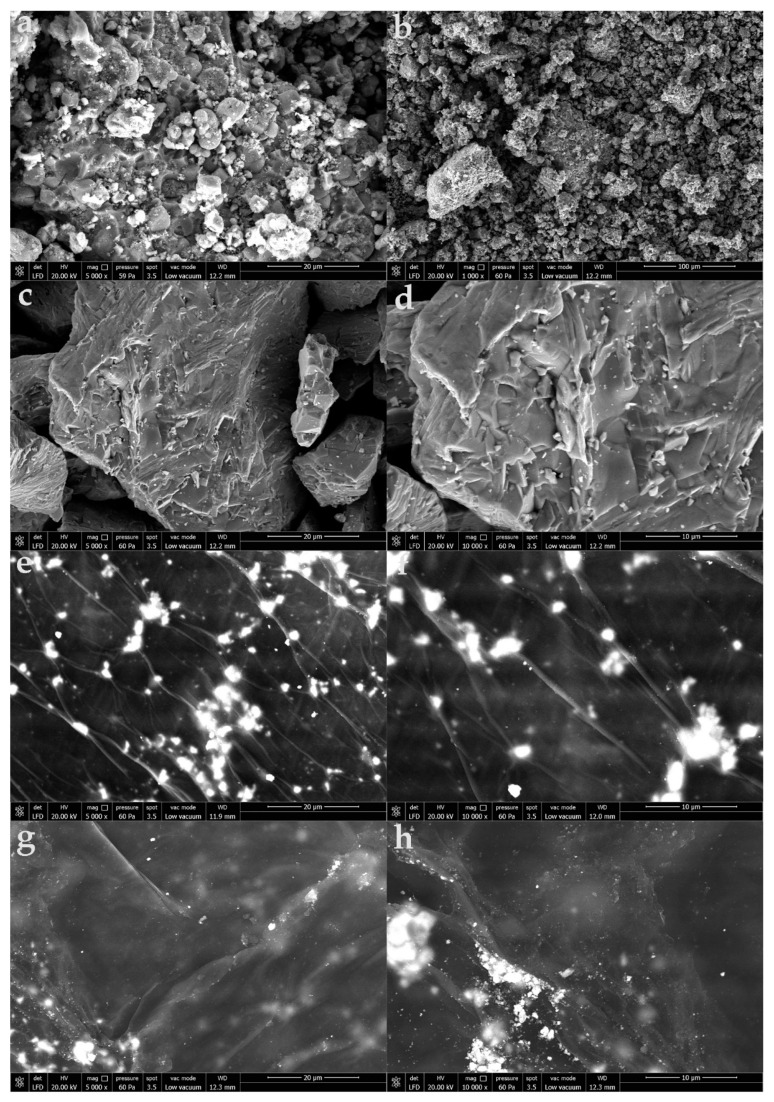
SEM images of WO_3_ (**a**,**b**), WC (**c**,**d**), WO_3_/GO ex situ synthesis (**e**,**f**), and WO_3_/GO in situ synthesis (**g**,**h**).

**Figure 6 materials-15-08019-f006:**
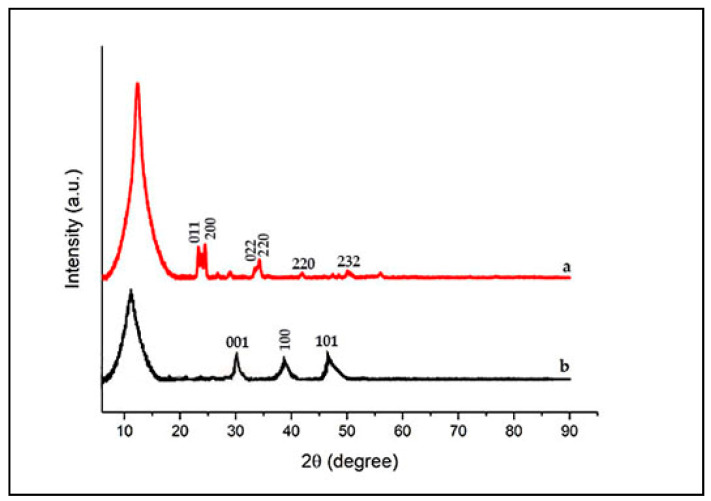
XRD patterns of WO_3_/GO ex situ synthesis (a) and WO_3_/GO in situ synthesis (b).

**Figure 7 materials-15-08019-f007:**
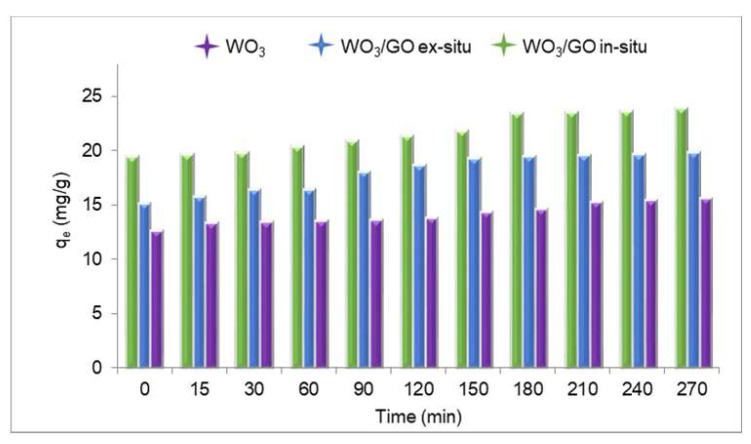
Relationship between adsorption capacity and time.

**Figure 8 materials-15-08019-f008:**
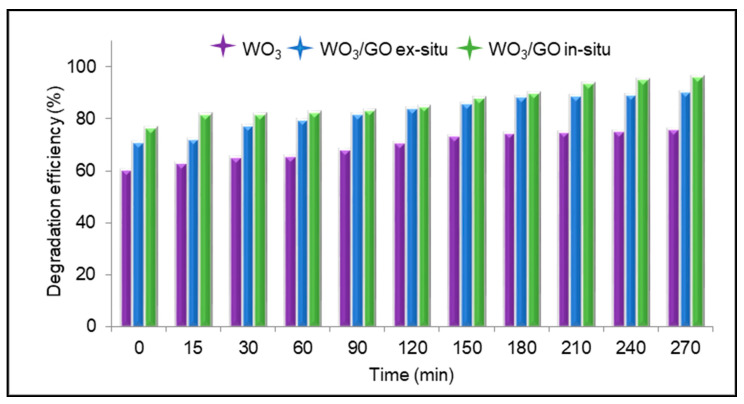
Degradation efficiency of MB.

**Figure 9 materials-15-08019-f009:**
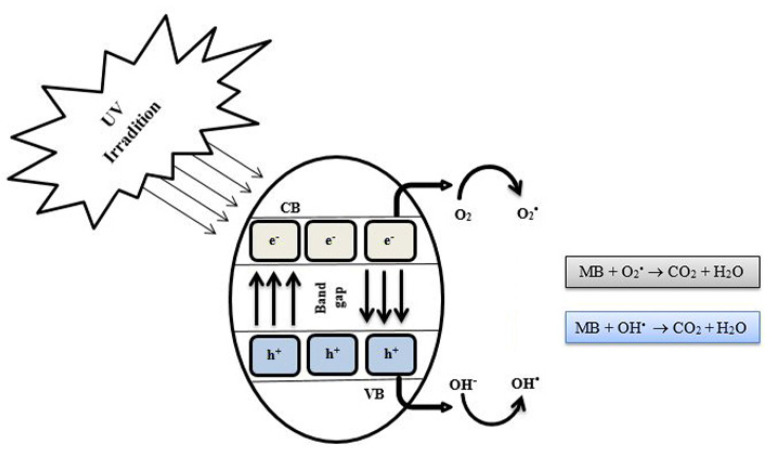
Schematic representation of photocatalytic degradation of methylene blue by WO_3_/GO nanocomposites under UV irradiation illumination.

**Table 1 materials-15-08019-t001:** Elemental composition of WO_3_, WC and WO_3_/GO in ex situ synthesis and in WO_3_/GO in situ synthesis.

Samples	% at.		
	Oxygen	Tungsten	Carbon
WO_3_	20.65	79.35	-
WC	-	90.34	9.66
WO_3_/GO ex situ	41.10	8.62	50.28
WO_3_/GO in situ	26.97	16.71	56.32

**Table 2 materials-15-08019-t002:** Comparison of photocatalytic performance between this work and reported references.

Photocatalyst	Methods of Synthesis	Photodecomposition	Photocatalytic Effciency	References
WO_3_/GO	In situ ex situ chemical oxidation	MB	96.30% 90.52%	Current work
WO_3_/GO	Ultrasonication Method	MB	97.03%	[28]
WO_3_/GO	Sol-gel method	MB	82%	[31]
WO_3_/GO	Photo-reduction method	MO	92.7%	[32]
WO_3_/GR	Hydrothermal method	MB	83%	[39]
WO_3_/rGO	In situ slvothermal method	MB	94%	[41]

## Data Availability

Not applicable.

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
