# Peer review of "The Effect of Different Morphologies of WO3/GO Nanocomposite on Photocatalytic Performance"

_materials, 2022, doi:10.3390/ma15228019_

Round 1
Reviewer 1 Report
There are many works to be done before the article can be published.
1. The abstract needs to be rewritten. There are no specific experimental results and conclusions in the current version.
2. The XRD patterns of WO₃, WO₃/GO should be added.
3. There is no evidence of WO₃/GO composite formation. HRTEM should be added.
4. Since the author talks about the influence of morphology on specific surface area, it is necessary to add specific surface area test of samples to confirm this statement.
5. The recombination of electrons and holes needs to be studied, and PL spectral test or electrochemical test must be added.
Author Response
First of all, we would like to thank the Reviewer for his/her improving remarks and the time spent for them.
Reviewer #1 comments
- The abstract needs to be rewritten. There are no specific experimental results and conclusions in the current version.
Abstract is revised. Experimental results and conclusions are included
- The XRD patterns of WO₃, WO₃/GO should be added.
EDS analyses and XRD spectrum of WO3/GO and the following sentences has been added in the results part.
- There is no evidence of WO₃/GO composite formation. HRTEM should be added.
EDS analyses and XRD spectrum of WO3/GO and the following sentences has been added in the results part.
The EDS analysis of WO3 powder shows a tungsten atomic percentage of 79.35 % and an oxygen atomic percentage of 20.65 % . WC powder shows tungsten 90.34 % and carbon atomic percentage of 9.66 % . After the synthesis of nanocomposites, a corresponding decrease in the atomic percentages of W element was observed. Furthermore, the presence of GO was confirmed via EDS analysis which showed the additional carbon elements and oxygen element WO3/GO ex-situ and in-situ, respectively (Table 1).
- Since the author talks about the influence of morphology on specific surface area, it is necessary to add specific surface area test of samples to confirm this statement.
In our study, the effect of WO3/GO nanocomposite structures synthesized by different methods on photocatalytic performance was investigated. When the synthesis results were examined by SEM, it was observed that different synthesis methods created different morphologies, and it was observed that photocatalytic degradation was also effective in these morphologies.
- The recombination of electrons and holes needs to be studied, and PL spectral test or electrochemical test must be added.
Physical mechanism of composite system was added to photocatalytic degradation results.

Reviewer 2 Report
In this article, the authors have synthesized a compound of tungsten oxide and graphene oxide by two different methods. In this, its photocatalytic capacity is analyzed against the degradation of methylene blue, the variations observed in this degradation are justified by the different morphology of the compounds. First of all, the error bars should have been included in Figure 6, to see if there are significant differences in the degradation of methylene blue. This degradation is justified by the morphology of the compounds, which is not clear in the SEM images, the different dispersion of WO3 in the GO being more likely. Characterization of these by EELS or EDS is necessary. The poor characterization of these composites, IR and SEM, in comparison with the precursor compounds, EDS and XPS spectra stands out.
I do not feel that this article is the special focus of quality for publication in this journal.
Author Response
First of all, we would like to thank the Reviewer for his/her improving remarks and the time spent for them.
Reviewer #2 comments
In this article, the authors have synthesized a compound of tungsten oxide and graphene oxide by two different methods. In this, its photocatalytic capacity is analyzed against the degradation of methylene blue, the variations observed in this degradation are justified by the different morphology of the compounds. First of all, the error bars should have been included in Figure 6, to see if there are significant differences in the degradation of methylene blue. This degradation is justified by the morphology of the compounds, which is not clear in the SEM images, the different dispersion of WO3 in the GO being more likely. Characterization of these by EELS or EDS is necessary. The poor characterization of these composites, IR and SEM, in comparison with the precursor compounds, EDS and XPS spectra stands out.
I do not feel that this article is the special focus of quality for publication in this journal.
In our study, the photocatalytic efficiency of the nanocomposite structure formed by both ex-situ and in-situ syntheses of WO3/GO composites was evaluated. WO3/GO nanocomposites with different morphological structures were synthesized in different synthesis methods. Photocatalytic effect against the degradation of methylene blue was observed in both synthesis methods. The results of EDS analyzes of nanocomposite structures were added to the article and revised.

Reviewer 3 Report
In this work, the authors used graphite and tungsten carbide (WC) in situ synthesis, and WO3 and graphene oxide in situ synthesis of tungsten oxide/graphene oxide (WO3/GO) nanocomposites with different morphologies. GO is synthesized by improved Hummer method. XRD, EDS and FTIR will be used to characterize nanocomposites. The photocatalytic activity of WO3/GO film will be evaluated by methylene blue (MB) adsorption and visible light photocatalytic degradation activity through ultraviolet visible spectrum. I believe that publication of the manuscript may be considered only after the following issues have been resolved.
1. In order to better highlight the advantages of this work, the author needs to provide a table to compare related work.
2. The authors need to rewrite the abstract. The introduction of background needs to be greatly reduced to highlight this work and its relevant significance.
3. In Fig. 6 and Fig. 7, it is suggested that the author supplement the corresponding degradation spectra.
4. The introduction can be improved. The articles related to the some applications of graphene materials should be added such as Sensors 2022, 22, 6483; ACS Sustain. Chem. Eng. 2015, 3, 1677–1685; RSC Adv. 2022, 12, 7821–7829; Talanta 2015, 134, 435–442.
5. What is the physical mechanism of such excellent performance of the composite system?
6. The text information in Figure 3, Figure 5 is not clear.
7. The existence of graphene in the composite system needs to be proved, such as Raman spectroscopy.
Author Response
First of all, we would like to thank the Reviewer for his/her improving remarks and the time spent for them.
- In order to better highlight the advantages of this work, the author needs to provide a table to compare related work.
Table is created and added to the article to compare related studies.
- In Fig. 6 and Fig. 7, it is suggested that the author supplement the corresponding degradation spectra.
İleri ki zamanlarda planlanan çalışmalarda bozunma spektrumları belli zaman aralıklarında deÄŸil bozunma tamamlananan kadar devam ettirilecektir.
Further investigations should evaluate the spectrums until degradation finalizes not in specific time intervals.
.4. The introduction can be improved. The articles related to the some applications of graphene materials should be added such as Sensors 2022, 22, 6483; ACS Sustain. Chem. Eng. 2015, 3, 1677–1685; RSC Adv. 2022, 12, 7821–7829; Talanta 2015, 134, 435–442.
Kindly refer papers below as they are highly relevant to this report:
The references were added to relevant section.
.
- What is the physical mechanism of such excellent performance of the composite system?
Physical mechanism of composite system was added to photocatalytic degradation results
- The text information in Figure 3, Figure 5 is not clear.
Fig 3 and 5 revised and enhanced the quality
- The existence of graphene in the composite system needs to be proved, such as Raman spectroscopy.
, EDS analyses and XRD spectrum of WO3/GO and the following sentences has been added in the results part.
The EDS analysis of WO3 powder shows a tungsten atomic percentage of 79.35 % and an oxygen atomic percentage of 20.65 %. WC powder shows tungsten 90.34 % and carbon atomic percentage of 9.66 %. After the synthesis of nanocomposites, a corresponding decrease in the atomic percentages of W element was observed. Furthermore, the presence of GO was confirmed via EDS analysis which showed the additional carbon elements and oxygen element WO3/GO ex-situ and in-situ, respectively (Table 1).

Reviewer 4 Report
Decision:
Minor revision
Comments
The authors reported the Effect of preheating temperature on thermal-mechanical properties of dry-vibrating MgO-based material lining in the tundish. Overall, the work is good and well-presented. However, the authors should address the following points outlined below to improve scientific quality. After the suggested revisions are carefully addressed, this work may be considered for publication
1. The abstract is too long. Make it short with the important highlights of this work. It should be clear and informative with important highlights and the main aim of the work. Lines 14-19 should be removed and added to the introduction section.
2. In the introduction, part authors should highlight the novelty of this work.
3. From lines 33 to 44 first part of the introduction section 15 ref was cited and most of them are very old the latest one if 2019. Cite new work related to this article and compare the result.
Antioxidants 2022, 11(6), 1064; https://doi.org/10.3390/antiox11061064
Crystals 2021, 11(12), 1467; https://doi.org/10.3390/cryst11121467
4. In the experimental part, mentioned the name and location of companies for each material and equipment used in this experiment
5. There are many grammatical errors, recheck the manuscript once again for all typo errors.
6. In figure 5 its hard to see the scale bar on SEM images
Author Response
First of all, we would like to thank the Reviewer for his/her improving remarks and the time spent for them.
- The abstract is too long. Make it short with the important highlights of this work. It should be clear and informative with important highlights and the main aim of the work. Lines 14-19 should be removed and added to the introduction section.
Line 14-19 is removed from abstract and added to introduction section. Introduction and abstract is revised as suggested.
- In the introduction, part authors should highlight the novelty of this work.
- From lines 33 to 44 first part of the introduction section 15 ref was cited and most of them are very old the latest one if 2019. Cite new work related to this article and compare the result.
Antioxidants 2022, 11(6), 1064; https://doi.org/10.3390/antiox11061064
Crystals 2021, 11(12), 1467; https://doi.org/10.3390/cryst11121467
Kindly refer papers below as they are highly relevant to this report:
Update references are replaced.
- In the experimental part, mentioned the name and location of companies for each material and equipment used in this experiment
The following sentences has been added in the results part.
Characterization WO3/GO Nanocomposite
The GO characterization was performed with X-ray diffraction (XRD) and Scanning Electron Microscopy (SEM/EDX) technique. WO3/GO composite were also evaluated by using a scanning electron microscope (SEM, Quanta Feg 250; FEI, Eindhoven, the Netherlands). WO3/GO composites were examined with a low-vacuum at 20.00 kV and 12.7–13.2 mm working distance, at 5000X and 10000X magnifications respectively. Elemental analysis of nanocomposites carried out using an SEM microscope equipped with an energy-dispersive X-ray spectroscopy (EDX, Quanta Feg 250; FEI, Eindhoven, the Netherlands). The distribution and atomic composition of WO3/GO was examined using elemental mapping at an accelerating voltage of 20 kV. The crystalline phase of WO3/GO were examined by X-ray diffraction (XRD, Bruker D8 Advance Twin-Twin; Bruker, Karlsruhe, Germany). at 40 kV, 40 mA, and 1600 watts. In order to determine the photocatalytic capacity of WO3/GO nanocomposites, UV-Vis Spectrophotometer (UV-Vis Carry 60) device was used.
- There are many grammatical errors, recheck the manuscript once again for all typo errors.
The manuscript is revised according to grammar and type errors.
- In figure 5 its hard to see the scale bar on SEM images
All the figures are revised.

Round 2
Reviewer 1 Report
1. It is suggested that the phase corresponding to the diffraction peak should be marked on the XRD pattern.
2. It is suggested to add the energy band structure diagram of WO3/GO composite to clarify the transfer direction of photogenerated charges.
3. In the discussion of morphology and BET, it is suggested to add these documents.
(1) Journal of Materials Science: Materials in Electronics, 2021, 32, 21511-21524.
(2) Journal of Materials Science: Materials in Electronics, 2019, 30, 13826-13834.
Author Response
First of all, we would like to thank the Reviewer for his/her improving remarks and the time spent for them.
Phase corresponding to the diffraction peak has been marked on the XRD pattern.Schematic representation of photocatalytic degradation of methylene blue by WO3/GO nanocomposites has been added in the result part.

Reviewer 2 Report
The modifications made to the article by the authors have significantly improved its quality. Although I consider that it would be necessary to clarify the mechanism by which MB degradation occurs, adding a diagram of this process.
Author Response
First of all, we would like to thank the Reviewer for his/her improving remarks and the time spent for them.
Schematic representation of photocatalytic degradation of methylene blue by WO3/GO nanocomposites has been added in the result part.

Reviewer 3 Report
Accept in present form.
Author Response
We would like to thank the Reviewer for his/her improving remarks and the time spent for them.
